# Research on Risk-Averse Procurement Optimization of Emergency Supplies for Mine Thermodynamic Disasters

**Weimei Li [1,2] and Leifu Gao [2,\*]**

1    School of Business Administration, Liaoning Technical University, Huludao 125105, China
2    Institute for Optimization and Decision Analytics, Liaoning Technical University, Fuxin 123000, China
*    Correspondence: gaoleifu@163.com

**Abstract:** Reducing procurement risks to ensure the supply of emergency supplies is crucial for mitigating the losses caused by mine thermodynamic disasters. The risk preference of decision-makers and supply chain collaboration are the important aspects for this reductiom. In this study, a novel P-CVaR (Piecewise conditional risk value) distributionally robust optimization model is proposed to accurately assist the decision-makers' decision of risk preference for reducing procurement risks. Meanwhile, the role of cooperation between procurement and reserves are considered for the weakening procurement risks. A risk-averse bi-level optimization model is proposed to obtain the optimal procurement strategy. Furthermore, by applying the Lagrange duality theorem, the complexity of the bi-level optimization model is simplified then solved using a PSO algorithm. Using empirical analysis, it has been verified that the model presented in this paper serves as a valuable guideline for mine thermodynamic pre-disaster emergency material procurement strategies for the prevention of thermodynamic disasters.

**Keywords:** mine thermodynamic disasters; emergency material procurement; risk preferences; CVaR distributionally robust optimization; bi-level optimization

**MSC:** 90B50





## 1. Introduction

### 1.1. Background

Mine thermodynamic disasters are complex disasters related to heat release and transfer. According to relevant statistics, thermodynamic disasters accounted for 60.9% of the severe and major mine accidents that occurred in China from 2000 to 2021 [1]. In the case of mine thermodynamic disasters, a severe accident occurred at the Babao Coal Mine in Tong Hua, Jilin province, China, on 1 April 2013. This accident was triggered by coal spontaneous combustion and a gas explosion in the goaf area, followed by a secondary gas explosion during the sealing of the working face. It resulted in 53 fatalities and direct economic losses of CNY 47.089 million. On 27 September 2020, a major fire accident occurred at the Song Zao Coal Mine in Chongqing, resulting in 16 fatalities, 42 injuries, and direct economic losses of CNY 25.01 million [2]. It is therefore evident that mine thermodynamic disasters can result in significant casualties and economic losses. It is essential to procure emergency supplies before mine thermodynamic disasters to ensure the timely implementation of emergency response activities and prevent significant losses. An accurate procurement strategy pre-disaster can prevent significant losses caused by purchasing at higher prices to meet extreme demand after the disaster. However, the same type of disaster phenomenon may be caused by the superposition of different causative mechanisms in mine thermodynamic disasters. For example, fire can be caused by mine fires, gas combustion, or coal seam explosions. In this context, it is difficult to accurately estimate the emergency supplies demand for mine thermodynamic disasters using traditional methods that rely on factors such as disaster types and occurrence locations. More

serious procurement risks are included in the emergency supplies for mine thermodynamic disasters due to the enormous uncertainty in demand. These risks can exacerbate the fatalities and economic losses in mine thermodynamic disasters. The risk-averse procurement strategy of emergency supplies for mine thermodynamic disasters is a practical and meaningful research topic.

### 1.2. Relate Works

Currently, researchers have proposed various risk-averse methods for emergency procurement in different perspectives. Some scholars have introduced supply chain contracts, such as framework agreements and option contracts, into emergency supplies procurement management to enhance procurement flexibility and reduce procurement risks. Tian Jun et al. [3] developed an emergency procurement model based on capability options contracts and demonstrated that this approach effectively reduces physical inventory while mitigating the risk of stockouts. Zhang et al. [4] conducted research on emergency supplies procurement under joint reserve based on framework agreements. They established a nonlinear mathematical model to determine the optimal procurement reserve and physical storage quantities, aiming to ensure the fulfillment of emergency demands while reducing procurement costs. The emergency supplies procurement risks can be reduced by enhancing supply chain collaboration to improve the flexibility of emergency supplies provisioning. However, the demand of emergency supplies for mine thermodynamic disasters is complex and uncertain, and relying solely on supply chain collaboration cannot eliminate risks effectively. It is necessary to consider multiple factors comprehensively and take further risk-averse measures to ensure the reliability of procurement strategies.

In the study of risk-averse procurement strategies problems, the procurement strategies optimization considering risk preferences has been extensively studied over the past decade [5–8]. Cai Xin et al. [9] proposed optimal ordering decision models based on the M-CVaR risk metric under uncertain demand conditions. They analyzed the impact of suppliers' risk preferences on ordering decisions, using case studies to show that optimal ordering strategies differ across different types of risk preferences. Elham et al. [10] considered disruption risks and used conditional value at risk (CVaR) for the assessment of risk under specific scenario probabilities. They studied supplier selection and strategy problems of order allocation under risk-neutral and risk-averse attitudes, finding that diversifying suppliers is a viable method to mitigate disruption risks and that the attitude of the decision-maker towards risk plays a significant role in supplier selection and order quantity. In summary, optimizing procurement strategies in response to sudden risks is closely related to decision-makers' risk preferences. Different risk preferences lead to varying procurement strategy adjustments and play a significant role in supply chain management. Therefore, understanding decision-makers' risk preferences is of paramount importance to effectively optimize emergency procurement decisions for mine thermodynamic disasters.

CVaR is a classic method of measuring risk preference at a certain confidence level. By selecting an appropriate confidence level, decision-makers can measure and manage risks as per their preferences and risk tolerance [11]. Assessing risk preference using CVaR optimization models relies on the real tail information of random variables. However, due to the difficulty in accurately estimating the stochastic demand distribution of emergency supplies in mine thermodynamic disasters, there are significant challenges in directly using CVaR to characterize the risk preferences of procurement decision-makers. Thus, the CVaR distributionally robust optimization model offers a solution approach for solving such problems [12]. The method aims to approximate the true distribution of random variables by capturing the distribution uncertainty set. Uncertainty sets are a central element in distributionally robust optimization models, and moments and statistical distances are widely used methods for constructing uncertainty sets. Li et al. [13] propose that moments and statistical distances should be combined to construct uncertainty sets. It is indicated that the distributionally robust optimization model based on moments and distribution distances increased probability distribution information, and further reduced the conservatism

of uncertainty sets and improved out-of-sample performance. Cornilly [14] studied the robust boundary problem of distortion risk measures with respect to the known potential risk at the Kth moment condition. They found that using uncertainty sets constructed with higher-order moments can capture distribution tail characteristics, helping narrow the gap between worst-case risk values and actual values. Glasserma [15] conducted simulations and demonstrated that relative entropy can be used as an indicator for effectively quantifying the distribution model error of uncertain sets approaching the true distribution. Feng [16] pointed out that using Wasserstein distance as a substitute for relative entropy has shown significant effectiveness in characterizing the model errors of distribution uncertainty sets.

The demand distribution of emergency supplies for mine thermodynamic disasters exhibits characteristics of having a small-sample and heavy-tail due to the sudden nature of the disasters. At these conditions, the use of lower order integer moments to characterize the random distribution tail information of CVaR robust optimization models may introduce significant statistical errors [17]. In addition, the Wasserstein distance generates a significant tail distribution measurement bias [18,19]. These unfavorable factors will seriously affect the accuracy of the robust optimization model for CVaR distribution. The risk preference of decision-makers in emergency material procurement for mine thermodynamic disasters is difficult to accurately measure based on traditional CVaR distributionally robust optimization models.

### 1.3. Contribution

This paper proposes a risk-averse procurement strategy for mine thermodynamic pre-disaster. The main contributions of this paper are summarized as follows:

- The risk-averse procurement strategy based on the joint reserve framework is proposed in this paper, which considers the optimization of physical supplies and capacity before mine thermodynamic disasters.
- A new P-CVaR distributionally robust model is proposed to characterize decision-makers' risk preferences based on small-sample and heavy-tailed features of emergency supplies demand distributions for mine thermodynamic disasters.
- An emergency supplies procurement strategy bi-level optimization model is established considering decision-makers' risk preference to balance the procurement risks and supplier benefits.

The rest of this paper is organized as follows. In Section 2, the problem description and proposed emergency material procurement framework are provided. In addition, underlying assumptions and symbol description are presented. Section 3 develops a P-CVaR distributionally robust optimization for decision-makers' risk preference measurement, and an emergency material procurement risk-averse optimization model is established. In addition, the models are solved and analyzed. Section 4 verifies the effectiveness of the model through empirical analysis. Some conclusions and management inspirations are summarized in Section 5.

## 2. Problem Description

### 2.1. Procurement Framework

In this study, the mine procurement department simultaneously has three functions: Prior to the mine thermodynamic disasters, the procurement department predicts emergency material demand based on probability distribution theory and historical mine thermal disasters, and selects suitable suppliers for framework agreements to joint reserve. Then, the risk of emergency material procurement and the risk preference of decision-makers are measured. Finally, a procurement strategy is determined according to the decision-maker's risk preference. This strategy includes the optimal joint reserves that determine the framework suppliers and the corresponding optimal capacity quantity, in addition to determining the pre-disaster amounts of physical reserves that are stored in the mine warehouse. After the mine thermodynamic disasters occur, the emergency physical

supplies reserved by the mine warehouse are first used for the emergency rescue operations. Then, emergency supplies are continuously transported from the framework agreement providers to the distribution center; then, the capacity is immediately produced and distributed to the disaster site. The emergency material procurement system described above is represented in Figure 1.

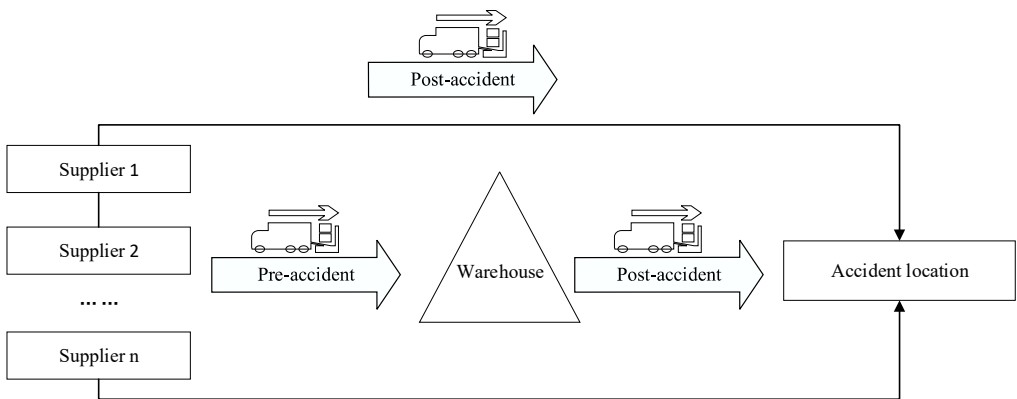

**Figure 1.** Emergency material procurement system for mine thermodynamic disasters.

The principle of emergency management for mine thermodynamic disasters is the prioritization of life. When coal mining enterprises make emergency material procurement decisions, the reliability of emergency material supply should be emphasized to reduce disaster losses. At the same time, supplier benefits should also be considered to ensure the reliability of the supply. Considering the uncertainty of the distribution of emergency material demand for mine thermodynamic disasters, a risk-averse bi-level distributionally robust optimization model is established for emergency material procurement. The upper level of the model aims to minimize the risk of coal mining enterprise procurement costs according to the decision-makers' risk preference, while the lower level of the model aims to minimize the production cost of suppliers. The robustness of the optimal procurement plan is ensured through the distributed robust optimization method in the case of uncertain demand distribution for thermal and dynamic disasters in mines. The interruption of emergency supplies supply is reduced through joint reserves. The reduction in procurement cost risk in coal mines and the increase in supplier profits are both achieved through a risk-averse bi-level optimization model based on P-CVaR.

### 2.2. Model Assumptions

Considering the suddenness characteristic of the mine thermodynamic disasters, the following assumptions are made:

1.  The sample of historical emergency material demand can reflect the distribution pattern of the demand variable for mine thermodynamic disasters.
2.  The random demand variable follows a heavy-tailed distribution with small-samples.
3.  Each supplier has an ample supply of emergency supplies.
4.  The emergency management department has the authority to determine the procurement strategy for emergency supplies.

### 2.3. Symbol Description

The symbols used in the model construction are as follows (Table 1).

**Table 1.** Symbol description.

| Parametric Variable | |
|---|---|
| $n$ | The set of candidate suppliers, $i \in n$ |
| $B$ | Maximum capacity per unit vehicle |
| $\gamma$ | Maximum capacity of coal mine warehouse |
| $\omega$ | The budget upper limit for emergency material procurement |
| $c^s$ | Unit inventory cost of physical supplies |
| $\tau^o$ | If the demand is less than the physical reserve, $\tau^o$ is 1; otherwise, it is 0 |
| $\tau^v$ | If the demand is between physical reserves and total reserves, the value is 1, otherwise it is 0 |
| $\tau^q$ | If the demand is greater than total reserves, it is 1; otherwise, it is 0 |
| $e^o$ | Unit residual value of expired physical supplies |
| $x$ | Emergency material demand for mine thermodynamic disasters |
| $d = (d_1, \ldots, d_n)$ | The distance from the supplier to the accident coal mine |
| $T = \{T_1, \ldots, T_n\}$ | Unit cost of the supplier transporting supplies to the accident coal mine |
| $c^p = \left\{c_1^p, \ldots, c_n^p\right\}$ | The selling unit price of physical supplies provided by the supplier |
| $c^r = \left\{c_1^r, \ldots, c_n^r\right\}$ | Reserve unit price of production capacity |
| $c^v = \left\{c_1^v, \ldots, c_n^v\right\}$ | The selling unit price of physical supplies produced by production capacity |
| $c^q = \left\{c_1^q, \ldots, c_n^q\right\}$ | Shortage unit cost due to insufficient reserves |
| $c^u = \left\{c_1^u, \ldots, c_n^u\right\}$ | Unit cost vector of additional physical supplies produced by suppliers |
| $\phi^u = \left\{\phi_1^u, \ldots, \phi_n^u\right\}$ | Unit price vector of additional physical supplies produced by suppliers |
| $\phi^v = \left\{\phi_1^v, \ldots, \phi_n^v\right\}$ | Unit cost of production capacity into supplies |
| $\phi^p = \left\{\phi_1^p, \ldots, \phi_n^p\right\}$ | The unit cost of physical supplies produced by suppliers |
| $\phi^r = \left\{\phi_1^r, \ldots, \phi_n^r\right\}$ | Unit cost reserved by supplier's production capacity |
| $\varphi^p = \left\{\varphi_1^p, \ldots, \varphi_n^p\right\}$ | The minimum physical procurement quantity allowed by the supplier |
| $\varphi^r = \left\{\varphi_1^r, \ldots, \varphi_n^r\right\}$ | Minimum capacity reserve allowed by the supplier. |
| $\psi^p = \left\{\psi_1^p, \ldots, \psi_n^p\right\}$ | Maximum physical supply quantity that the supplier can provide |
| $\psi^r = \left\{\psi_1^r, \ldots, \psi_n^r\right\}$ | The maximum capacity reserve that the supplier can provide |
| Decision variables | |
| $y = (y_1, \ldots y_n)$ | Physical reserves quantity purchased from supplier $i$ |
| $z = (z_1, \ldots z_n)$ | Capacity reserve quantity purchased from supplier $i$ |

## 3. Model Construction

In the procurement system for joint reserves based on the framework agreement, the cost of coal mines as purchasers within the unit reserve cycle is as follows:

$$C(y,z) = \sum_{i=1}^{n} \left(c_i^p y_i + c^s y_i + c_i^r z_i + T_i d_i \frac{y_i}{B}\right) - \sum_{i=1}^{n} (\tau^o e^o y_i) + \sum_{i=1}^{n} \left(\tau^u \left(c_i^v (x - y_i) + T_i d_i \frac{(x - y_i)}{B}\right)\right) + \sum_{i=1}^{n} \left(\tau^v \left(c_i^v z_i + T_i d_i \frac{z_i}{B} + c^q (x - y_i - z_i)\right)\right) \tag{1}$$

The first part of the above formula is the procurement cost, including the physical procurement cost, physical inventory cost, capacity storage cost, and transportation cost; the second part is the residual value profit of residual assets. The third part is the cost of purchasing physical supplies produced by production capacity and the cost of transporting these supplies. The fourth part is the cost of purchasing physical supplies produced by production capacity and the cost of transporting these supplies, as well as the cost of purchasing additional emergency supplies to meet actual demand.

The cost per unit reserve cycle for suppliers is as follows:

$$W(y,z) = \sum_{i=1}^{n} \left(\phi_i^p y_i + \phi^r z_i - c_i^p y_i - c_i^r z_i\right) + \sum_{i=1}^{n} \left(\tau^u \left(\phi_i^u (x - y_i) - c_i^u (x - y_i)\right)\right) + \sum_{i=1}^{n} \left(\tau^v \left(\phi_i^u z_i - c_i^u z_i\right)\right) \tag{2}$$

The first part of the above formula is the production cost of the supplier, including the cost of physical emergency material production and reserve capacity. The second part is the cost of producing a portion of the production capacity into physical goods; the third part is the cost of producing all production capacity into physical products.

*3.1. Risk Preference Measured Optimization Model*

The decision-makers' risk preference measurement model is the main content in the risk-averse procurement optimization of emergency supplies for mine thermodynamic disasters. In this paper, the demand for emergency supplies is evaluated by its probability distribution. The risk preferences are measured via a CVaR distributionally robust optimization model. In the following, the classical model of a CVaR distributionally robust optimization with moments and distribution distance is first introduced, and then the P-CVaR model is proposed based on this to accurately describe the risk preference of emergency material procurement decision-makers for mine thermodynamic disasters.

3.1.1. CVaR Distributionally Robust Optimization

The classical CVaR distributionally robust optimization model based on moments and distribution distances can be described in the following form [20]:

$$\max_{P \in T_0} \mathrm{CVaR}_\alpha(l(x)) \tag{3}$$

$$\mathrm{CVaR}_\alpha[l(x)] = \min_\beta \{\beta + \frac{1}{1-\alpha} E_p[(l(x) - \beta)^+]\} \tag{4}$$

where $x = \{x_1, \ldots, x_N\}$ is a random variable with an unknown probability distribution $p$ and a density function $g(x)$; $l(x)$ is the loss function; $\beta$ denotes the expected loss; $\alpha$ represents the confidence level for controlling risk aversion. The unknown true probability distribution $p$ of variable $x$ is defined by the following uncertain set:

$$\Gamma_0 = \{p : E[x^\theta] = \mu, \frac{dp}{dq} \leq \eta\} \tag{5}$$

In the formula above, $E[x^\theta] = \mu$ is the $\theta$-th moments reflecting the true distribution characteristics; $\frac{dp}{dq} \leq \eta$ is a criterion that measures errors of the distribution model; $q \in P$ is the prior reference distribution of $x$ with a density function of $f(x)$.

The CVaR distributionally robust optimization model based on Wasserstein distance and second-order moment [13] is one of the classic models described in formula (3). Its expression is as follows

$$\max_{P \in \Gamma} \mathrm{CVaR}_\alpha(l(x)) \tag{6}$$

$$\Gamma = \left\{ p : E[x^2] = \sigma, W(p,q) \leq \eta \right\} \tag{7}$$

where $E[x^2] = \mu$ is the second-order moment of distribution $p$. $W(p,q)$ is the Wasserstein distance between the probability distributions $p$ and $q$.

$$W(p,q) = \inf_{\gamma \in \Pi} \sum_{i=1}^N \sum_{j=1}^N c(x_i, \xi_j) \gamma(x_i, \xi_j) \tag{8}$$

$\xi \sim q$, $c(x, \xi) : S \times S \to R$ is the cost function, $\gamma(x, \xi) \in \Pi$ is a joint distribution of $p$ and $q$, and satisfies

$$\begin{cases} \sum\limits_{i=1}^N \gamma(x_i, \xi_j) = q_j, i,j = 1, \ldots N, \\ \sum\limits_{j=1}^N \gamma(x_i, \xi_j) = p_i, i,j = 1, \ldots, N, \\ \sum\limits_{i=1}^N \sum\limits_{j=1}^N \gamma(x_i, \xi_j) = 1, i,j = 1, \ldots N. \end{cases} \tag{9}$$

The classic CVaR distributionally robust optimization model (6) can accurately measure the risk preference of decision-makers in large sample sizes.

**Definition 1.** *Fractional moments [21]*

$$E[x^\theta] = \int_S x^\theta f(x)dx, \theta \in R \tag{10}$$

*It should be indicated that fractional moments not only accurately capture distribution information more than integer moments under a small-sample, but also have outstanding advantages in capturing tail characteristics of heavy-tail distributions.*

**Definition 2.** *Wasserstein distance with entropy constraint [22,23].*

$$\begin{cases} W(p,q) = \inf_{\gamma \in \Pi} \sum_{i=1}^N \sum_{j=1}^N c(x_i, \xi_j)\gamma(x_i, \xi_j) \\ H(p,q) = \sum_{i=1}^N \sum_{j=1}^N -\gamma_l(x,\xi)\log(\gamma_l(x,\xi)) \geq \kappa \end{cases} \tag{11}$$

*Wasserstein distance can measure the distance between any two distributions. However, the Wasserstein distance suffers from the curse of dimensionality when computing high-dimensional sample data. Introducing information entropy constraints based on the Wasserstein distance can transform optimization problems containing the Wasserstein distance into strictly convex problems and more reasonably represent the uncertainty of distribution.*

3.1.2. P-CVaR Distributionally Robust Optimization Model

Mine thermodynamic disasters are generally sudden and major disasters. This makes its demand distribution of emergency supplies exhibit characteristics of being small-sample and heavy-tailed. At these conditions, the uncertain set $\Gamma$ of the classic CVaR distributionally robust optimization model (6) faces the challenge of accurately estimating the uncertainty of distribution. Two reasons for the bias are that second-order moments cannot capture the distribution tail features of heavy-tailed distributions in small-samples, and the Wasserstein distance can also lead to significant errors in measuring heavy-tailed distributions due to the different selection of cost functions. These may lead to a significant bias in using the classical CVaR model to measure the risk preferences of procurement decision-makers in mine thermodynamic disasters. In order to accurately measure the risk preference of decision-makers in emergency material procurement for mine thermodynamic disasters, a P-CVaR distributionally robust optimization model will be proposed based on fractional moments, Wasserstein distance with entropy constraint, and piecewise optimization.

Dividing the distribution support set S of the random demand variable $x$ into $L$ sub-support sets $S_l, l = 1, \ldots, L$, each of which satisfy the following conditions:

$$S = \bigcup_{l=1,\ldots,L} S_l \tag{12}$$

$$l_1 \neq l_2 \Rightarrow S_{l_1} \cap S_{l_2} = \phi \tag{13}$$

On each sub-support set $S_l$, the P-CVaR optimization model based on negative fractional moments and Wasserstein distance is established. The P-CVaR distributionally robust optimization model can be expressed in the following form based on the total probability principle.

$$\text{P-CVaR}_\alpha(l(x,y)) = \sum_{l \in L} \overline{p}_l (\max_{P_l \in \Gamma_l} \text{CVaR}_\alpha(l(x,y))) \tag{14}$$

$$\Gamma_l = \begin{cases} \int\limits_{S_l \times S_l} (x)^{\theta_l} \gamma_l(x,\xi) dx d\xi = \mu_l \\ \int\limits_{S_l \times S_l} c_l(x,\xi) \gamma_l(x,\xi) dx d\xi \leq \eta_l \\ \int\limits_{S_l \times S_l} -\gamma_l(x,\xi) \log(\gamma_l(x,\xi)) dx d\xi \geq \kappa_l \\ \int\limits_{S_l \times S_l} \gamma_l(x,\xi) dx d\xi = \dfrac{1}{\overline{p}_l} \end{cases} \tag{15}$$

where the $\overline{p}_l = prop\{x_l \in S_l\}, l = 1, \ldots, L$ and can be calculated from sample data.

In the P-CvaR model, the fractional moment increases the constraint that can more finely characterize the tails of heavy-tailed distributions, and the selection of the Wasserstein distance cost function improves the approximation effect of the CVaR risk measurement problem. The accuracy of measuring risk preference using the P-CVaR distributionally robust optimization model has been improved through the piecewise method. This accuracy is crucial to risk-averse emergency material procurement for mine thermodynamic disasters. The P-CVaR model overcomes the limitations of Wasserstein distance in measuring differences in distribution tails and achieves the control and optimization of tail distribution and overall model errors.

### 3.2. Risk-Averse Procurement Optimization Model

Based on the above P-CVaR model, a risk-averse bi-level distributionally robust optimization model for emergency material procurement can be expressed as follows:

$$\begin{cases} \begin{cases} \min\limits_{y} \max\limits_{P \in \Gamma_l} \text{P-CVaR}_\alpha(C(y,z)) \\ \text{s.t.} \sum\limits_{i=1}^{n} c_i^p y_i + c^s y_i + c_i^r z_i \leq \varpi \\ \sum\limits_{i=1}^{n} y_i \leq \gamma \\ \varphi_i^p \leq y_i \leq \psi_i^p \end{cases} \\ \begin{cases} \min\limits_{z} \max\limits_{P \in \Gamma_l} E(W(y,z)) \\ \text{s.t.} \varphi_i^r \leq z_i \leq \psi_i^r \end{cases} \end{cases} \tag{16}$$

In the above model, the upper-level objective function accurately reduces the risk of extreme loss in procurement costs by introducing P-CVaR. The first constraint indicates that the procurement cost cannot exceed the budget of the coal mine. The second constraint is that the physical procurement quantity does not exceed the maximum inventory level of the coal mine. The third constraint represents the physical procurement quantity meeting the minimum supply quantity of the supplier and that it cannot exceed its maximum supply quantity. The lower objective function is to ensure optimal profits for suppliers by reducing their production costs. The constraints of the lower optimization model indicate that the production capacity reserve must be greater than the supplier's minimum production capacity reserve and cannot exceed its maximum production capacity reserve.

In the robust optimization model of emergency material procurement distribution based on joint reserves, the robustness of the optimal procurement plan is ensured through the distributed robust optimization method in the case of uncertain demand distribution for thermal and dynamic disasters in mines. The interruption of emergency supplies supply is reduced through joint reserves. The reduction in procurement cost risk in coal mines and the increase in supplier profits are both achieved through a risk-averse bi-level optimization model based on P-CVaR.

### 3.3. Model Transformation and Solving

The objective function of the risk-averse bi-level distributionally robust optimization model (16) cannot be directly solved due to the unknown distribution $p$. It is necessary to

transform the dual theorem into a deterministic problem to reduce the complexity of the solution. The specific process is as follows.

The interior model of the upper-level objective function of the bi-level model (16) is

$$\max_{P \in \Gamma_l} \mathrm{CVaR}_\alpha(C(y,z)) \tag{17}$$

This can be rewritten in the following form based on the definition of CVaR and $\Gamma_l$.

$$\min_{\beta_l}\{\beta_l + \frac{1}{1-\alpha}\max_{\gamma_l}\{(\int_{S_l \times S_l}(C(y,z)-\beta_l)^+\gamma_l(x,\xi)dxd\xi\} \tag{18}$$

$$\text{s.t.} \begin{cases} \int_{S_l \times S_l}(x)^{\theta_l}\gamma_l(x,\xi)dxd\xi = \mu_l \\ \int_{S_l \times S_l}c_l(x,\xi)\gamma_l(x,\xi)dxd\xi \leq \eta_l \\ \int_{S_l \times S_l}-\gamma_l(x,\xi)\log(\gamma_l(x,\xi))dxd\xi \geq \kappa_l \\ \int_{S_l \times S_l}\gamma_l(x,\xi)dxd\xi = \frac{1}{p_l} \end{cases}$$

The internal maximization problem of the above model can be organized as follows

$$\max_{\gamma_l}\{(\int_{S_l \times S_l}(C(y,z)-\beta_l)^+\gamma_l(x,\xi)dxd\xi\}$$

$$\text{s.t.} \begin{cases} \int_{S_l \times S_l}(x)^{\theta_l}\gamma_l(x,\xi)dxd\xi = \mu_l \\ \int_{S_l \times S_l}c_l(x,\xi)\gamma_l(x,\xi)dxd\xi \leq \eta_l \\ \int_{S_l \times S_l}-\gamma_l(x,\xi)\log(\gamma_l(x,\xi))dxd\xi \geq \kappa_l \\ \int_{S_l \times S_l}\gamma_l(x,\xi)dxd\xi = \frac{1}{p_l} \end{cases} \tag{19}$$

This is a convex optimization problem about $\gamma_l(x,\xi)$ that is continuously differentiable and satisfies the constraint gauge condition and strong duality theorem. It can be solved according to the Lagrange duality theorem [24]. The Lagrange dual function of the model is

$$\min_{\lambda_{l1},\lambda_{l2},\lambda_{l3},\lambda_{l4}} \mu_l\lambda_{l1} + \lambda_{l2}\eta_l - \lambda_{l3}\kappa_l + \frac{1}{p_l}\lambda_{l4} +$$
$$\max_{\gamma_l(x|y)} \int_{S_l}\gamma_l(x|\xi)((C(y,z)-\beta_l)^+ - \lambda_{l1}x^{\theta_l} - \lambda_{l2}c_l(x,\xi) - \lambda_{l3}\log\gamma_l(x|\xi) - \lambda_{l4})dx \tag{20}$$

$\lambda_{l1}$, $\lambda_{l2}$, $\lambda_{l3}$, $\lambda_{l4}$ are Lagrange multipliers. According to the duality theory, model (20) can be transformed into the following model:

$$\begin{cases} \min_{\lambda_{l1},\lambda_{l2},\lambda_{l3},\lambda_{l4}} \mu_l\lambda_{l1} + \lambda_{l2}\eta_l - \lambda_{l3}\kappa_l + \frac{1}{p_l}\lambda_{l4} \\ \text{s.t. } (C(y,z)-\beta_l)^+ - \lambda_{l1}x^{\theta_l} - \lambda_{l2}c_l(x,\xi) - \lambda_{l3}\log\gamma_l(x|\xi) - \lambda_{l4} - \lambda_{l3} = 0 \\ \lambda_{l1},\lambda_{l4} \in R, \lambda_{l2},\lambda_{l3} \geq 0 \end{cases} \tag{21}$$

Therefore, the upper-level model of the bi-level model (16) can be equivalently written as

$$\min_{\lambda_{l1},\lambda_{l2},\lambda_{l3},\theta_l,\beta_l} \beta_l + \frac{1}{1-\alpha}(\mu_l\lambda_{l1} + \lambda_{l2}\eta_l - \lambda_{l3}\kappa_l + \frac{1}{p_l}\lambda_{l4})$$
$$\begin{cases} \text{s.t. } (C(y,z)-\beta_l)^+ - \lambda_{l1}x^{\theta_l} - \lambda_{l2}c_l(x,\xi) - \lambda_{l3}\log\gamma_l(x|\xi) - \lambda_{l4} - \lambda_{l3} = 0. \\ \lambda_{l1},\lambda_{l4} \in R, \lambda_{l2},\lambda_{l3} \geq 0. \end{cases} \tag{22}$$

In other words, the upper-level optimization model of the risk-averse bi-level optimization model (16) can be equivalently transformed into the following model.

Similarly, the lower-level model of the bi-level model (16) can be equivalently written as

$$
\begin{cases}
\sum_{l \in L} \overline{p}_l \left( \min_{z, \pi_{l1}, \pi_{l2}, \pi_{l3}, \pi_{l4}} (\pi_{l1} \mu_l + \pi_{l2} \eta_l - \pi_{l3} \kappa_l + \pi_{l4} \frac{1}{\overline{p}_l}) \right) \\
\text{s.t. } W(y, z) - \pi_{l1} x^{\theta_l} - \pi_{l2} c_l(x, y) - \pi_{l3} \log \gamma_l(x|y) - \pi_{l4} - \pi_{l3} = 0 \\
\varphi_i^r \leq z_i \leq \psi_i^r, \\
\pi_{l1}, \pi_{l4} \in R, \pi_{l2}, \pi_{l3} \geq 0
\end{cases}
\tag{23}
$$

$\pi_{l1}, \pi_{l2}, \pi_{l3}, \pi_{l4}$ are also Lagrange multipliers.

The final equivalent form of the risk-averse bi-level optimization model (16) can be obtained by organizing the equivalent upper-level optimization problem and lower-level optimization problem. The equivalent form is the following model (24):

$$
\begin{cases}
\begin{cases}
\sum_{l \in L} \overline{p}_l \left( \min_{y, \lambda_{l1}, \lambda_{l2}, \lambda_{l3}, \theta_l, \beta_l} \beta_l + \frac{1}{1 - \alpha} (\mu_l \lambda_{l1} + \lambda_{l2} \eta_l - \lambda_{l3} \kappa_l + \frac{1}{\overline{p}_l} \lambda_{l4}) \right) \\
\text{s.t. } (C^p(y, z) - \beta_l)^+ + \lambda_{l1} x^{\theta_l} - \lambda_{l2} c_l(x, \xi) - \frac{\lambda_{l3}(C(y, z) - \pi_{l1} x^{\theta_l} - \pi_{l2} c_l(x, y) - \pi_{l4} - \pi_{l3})}{\pi_{l3}} - \lambda_{l4} - \lambda_{l3} = 0 \\
\sum_{i=1}^n c_i^p y_i + c^s y_i + c_i^r z_i \leq \varpi, \\
\sum_{i=1}^n y_i \leq \gamma, \varphi_i^p \leq y_i \leq \psi_i^p, \\
\lambda_{l1}, \lambda_{l4} \in R, \lambda_{l2}, \lambda_{l3} \geq 0, \\
2 \leq \theta_l \leq 4, \theta_l \in R
\end{cases} \\
\begin{cases}
\sum_{l \in L} \overline{p}_l \left( \min_{z, \pi_{l1}, \pi_{l2}, \pi_{l3}, \pi_{l4}, \theta_l} (\pi_{l1} \mu_l + \pi_{l2} \eta_l - \pi_{l3} \kappa_l + \pi_{l4} \frac{1}{\overline{p}_l}) \right) \\
\text{s.t. } C(y, z) - \pi_{l1} x^{\theta_l} - \pi_{l2} c_l(x, y) - \pi_{l3} \frac{(W(y, z) - \beta_l)^+ + \lambda_{l1} x^{\theta_l} - \lambda_{l2} c_l(x, \xi) - \lambda_{l4} - \lambda_{l3}}{\lambda_{l3}} - \pi_{l4} - \pi_{l3} = 0 \\
\varphi_i^r \leq z_i \leq \psi_i^r. \\
\pi_{l1}, \pi_{l4} \in R, \pi_{l2}, \pi_{l3} \geq 0
\end{cases}
\end{cases}
\tag{24}
$$

The above transformed model is a deterministic and nonlinear bi-level programming problem [25]. This paper uses the PSO algorithm [26] to solve it.

## 4. Case Analyses

As a case study, this paper examines a major mine thermodynamic disaster incident that occurred at the Fuhua Coal Mine in Hegang City, China on 20 September 2008. The actual firefighting process consumed 825 tons of liquid $CO_2$, with three suppliers capable of providing liquid $CO_2$. The random demand for liquid carbon dioxide is treated as a random variable $x$. Using the sample data of liquid $CO_2$ demand from 39 historical accidents [27] (Table 2), the random liquid $CO_2$ demand probability distribution and procurement risks for the Fuhua mine thermodynamic disaster are evaluated based on sample data, and procurement strategies are established based on decision-maker risk preferences to determine the optimal physical procurement quantity and capacity substitute reserves.

**Table 2.** Demand data for liquid $CO_2$ in fire accidents.

| Accidents | Demand | Accident | Demand | Accidents | Demand |
|---|---|---|---|---|---|
| 1 | 30 tons | 14 | 160 tons | 27 | 20 tons |
| 2 | 220 tons | 15 | 40 tons | 28 | 380 tons |
| 3 | 190 tons | 16 | 23 tons | 29 | 15 tons |
| 4 | 170 tons | 17 | 36 tons | 30 | 27 tons |
| 5 | 420 tons | 18 | 440 tons | 31 | 40 tons |
| 6 | 450 tons | 19 | 320 tons | 32 | 240 tons |
| 7 | 360 tons | 20 | 260 tons | 33 | 270 tons |
| 8 | 45 tons | 21 | 225 tons | 34 | 480 tons |
| 9 | 25 tons | 22 | 520 tons | 35 | 45 tons |

**Table 2.** *Cont.*

| Accidents | Demand | Accident | Demand | Accidents | Demand |
|---|---|---|---|---|---|
| 10 | 35 tons | 23 | 540 tons | 36 | 105 tons |
| 11 | 180 tons | 24 | 825 tons | 37 | 36 tons |
| 12 | 50 tons | 25 | 32 tons | 38 | 740 tons |
| 13 | 205 tons | 26 | 55 tons | 39 | 500 tons |

*4.1. Hypothesis Testing*

In this section, it will be verified that the probability distribution of the random variable $x$ has small-sample and heavy-tailed distribution characteristics. According to the data in Table 2, the small-sample feature is obvious. Meanwhile, the statistical characteristics of the sample data of liquid $CO_2$ demand in Table 2 are analyzed, and the results are shown in Figure 2.

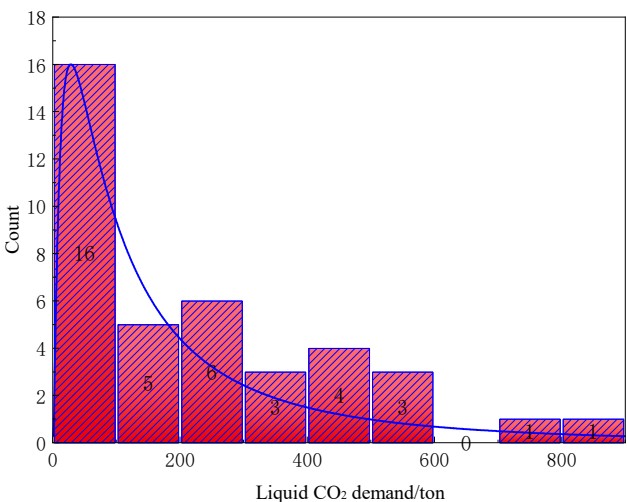

**Figure 2.** Histogram of sample data for liquid $CO_2$ demand.

Figure 2 shows the histogram distribution of demand sample data. It can be clearly seen that the demand distribution for liquid $CO_2$ is a positively skewed heavy-tailed distribution. The demand in the [700, 900] range can be considered as extreme demands, and the liquid $CO_2$ demand for the Fuhua coal mine accident is an extreme event.

Lognormal distribution, exponential distribution, and Weibull distribution are classic heavy-tailed distributions that can be used to describe the demand for emergency supplies. In order to further determine the type of random demand distribution for liquid $CO_2$, the distribution of the sample data for liquid $CO_2$ is fitted based on the principle of maximum likelihood. The fitting results are shown in Table 3 and Figure 3.

**Table 3.** Distribution goodness of liquid $CO_2$.

| Heavy-Tailed Distribution | Log Likelihood | Fitting Effect |
|---|---|---|
| Lognormal | −250.413 | best |
| Weibull | −250.125 | normal |
| Exponential | −250.134 | better |

Note: The smaller the Log likelihood value, the better the distribution for liquid $CO_2$ demand is fitted.

Observing and analyzing the fitting results shown in Table 3, the Log likelihood value obtained via lognormal distribution fitting is −250.413, which is the smallest compared to the −250.134 for the exponential distribution and −250.125 for the Weibull distribution. This result indicates that the liquid $CO_2$ demand for Fuhua coal mine follows a lognormal distribution.

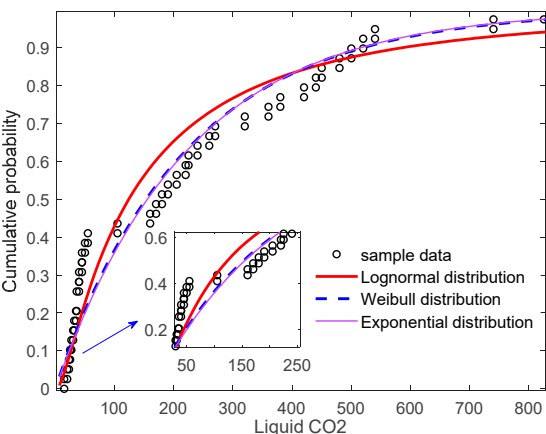

**Figure 3.** The fitted distribution of liquid $CO_2$ demand.

Figure 3 shows the fitted distribution of liquid $CO_2$ demand; it can be observed that compared to exponential and Weibull distribution, the tail of the distribution for liquid $CO_2$ demand can be more accurately represented by a lognormal distribution.

Based on the above results, the distribution of liquid $CO_2$ random demand for Fuhua coal mine has small-sample and heavy-tailed characteristics; assumption 2 is held. The distribution type is lognormal distribution.

### 4.2. Benefits of the Proposed Models

This section aims to demonstrate that the performance of the proposed risk-averse bi-level distributionally robust optimization model (16) for emergency material procurement under the liquid $CO_2$ random demand distribution is lognormal. Firstly, the fact that P-CVaR is more accurate in measuring decision-makers' risk preferences compared to classical CVaR models will be verified. Secondly, the impact of decision-makers' varying degrees of risk aversion preferences on the procurement model and the impact of this accuracy on emergency material procurement models will be explored. The parameter values used in the risk-averse bi-level optimization model are shown in Table 4.

**Table 4.** Parameter values.

| | | | |
|---|---|---|---|
| $c^p = (0.70, 0.70, 0.70)$ | $c^u = (0.50, 0.40, 0.30)$ | $\varphi^r = (100, 100, 100)$ | $d = (100, 150, 300)$ |
| $c^s = (0.20, 0.20, 0.20)$ | $\phi^u = (0.95, 0.95, 0.95)$ | $\psi^p = (100, 100, 100)$ | $B = (25, 25, 25)$ |
| $c^r = (0.40, 0.45, 0.50)$ | $\phi^p = (0.40, 0.30, 0.20)$ | $\psi^r = (500, 500, 500)$ | $\gamma = 300$ |
| $c^v = (0.70, 0.70, 0.70)$ | $\phi^r = (0.20, 1.50, 0.10)$ | $T = (0.15, 0.18, 0.20)$ | $\omega = 1000$ |
| $c^q = (1.00, 1.00, 1.00)$ | $\varphi^p = (80.0, 80.0, 80.0)$ | | |

#### 4.2.1. Benefits of the P-CVaR Models

In the progress of emergency material procurement for mine thermodynamic disasters, the analysis of the impact of piecewise number L and fractional moments on the accuracy of the decision-maker's risk preference measurement of the optimal procurement strategy for the risk-averse bi-level optimization model under the liquid $CO_2$ demand for follows a lognormal distribution. Three scenarios are set up as follows in the analyses.

Scenario 1: Risk-averse bi-level distributionally robust optimization model based on CVaR with second moment and Wasserstein distance.

Scenario 2: Risk-averse bi-level distributionally robust optimization model based on P-CVaR with fractional moments and piecewise number L = 1.

Scenario 3: Risk-averse bi-level distributionally robust optimization model based on P-CVaR with fractional moments and optimal piecewise number L = L* determined by multiple attempts.

In the scenarios above, the Scenario 2 models would be compared with the model of Scenario 1 to verify the fractional moments impact on the accuracy of decision-maker risk preference measurement in the risk-averse bi-level optimization model. Meanwhile, compared with the Scenario 2 model, the Scenario 3 model is set to verify the piecewise numbers impact on the accuracy of decision-maker risk preference measurement in the risk-averse bi-level optimization model.

We can solve the optimization models of the above scenarios via the PSO algorithm. In the obtained results, the worst-case CVaR is shown in Table 5, and the worst-case distribution of liquid $CO_2$ demand is as shown in Figure 4.

**Table 5.** The worst-case CvaR of the risk-averse bi-level optimization model for liquid $CO_2$ procurement.

| Scenarios | Piecewise Number (L) Lnumber Number | Worst-Case CVaR | True CVaR | Relative Error (%) |
|---|---|---|---|---|
| 1 | - | 85.5901 | 58.0546 | 47.4303 |
| 2 | 1 | 79.3031 | 58.0546 | 36.6008 |
| 3 | 6 | 60.7848 | 58.0546 | 4.7028 |

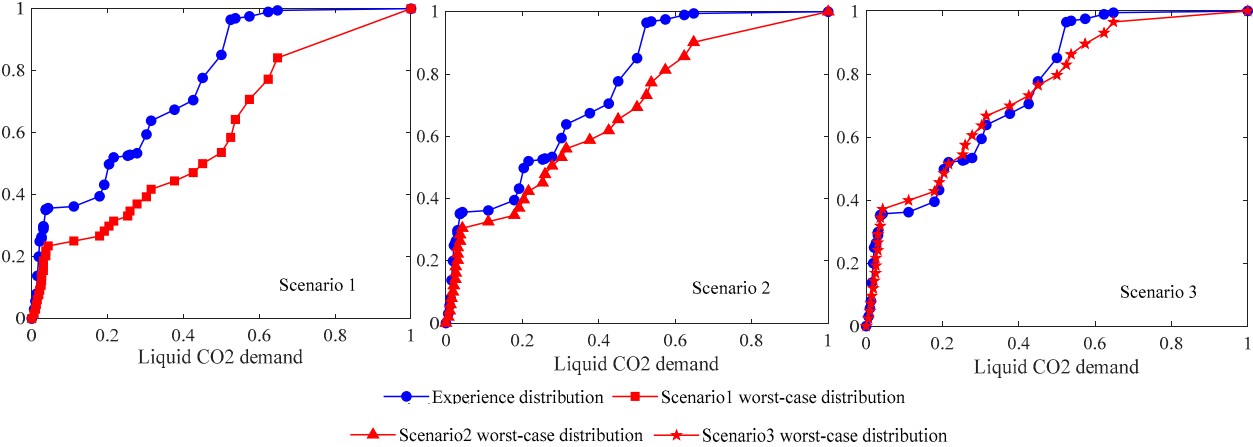

**Figure 4.** Worst-case distribution of the risk-averse bi-level optimization model for liquid $CO_2$ procurement.

The analysis of the worst-case risk of the risk-averse bi-level optimization model for emergency material procurement under different scenarios can be seen in Table 5. The relative error (36.6008) of Scenario 2 is smaller than the relative error (47.4303) of Scenario 1. This indicates that fractional moments are more effective in improving the accuracy of measuring risk preference compared to second-order integer moments. The relative error of Scenario 3 risk assessment is the smallest at 4.7028%, which means that using the piecewise Wasserstein distance can help improve the accuracy of measuring risk preference. The above content verifies that the P-CVR distributionally robust optimization model with fractional moments and an optimal piecewise number can provide a more accurate risk preference measurement of decision-makers for emergency material procurement.

Through the analysis of Figure 4, it can be observed that the worst-case distribution optimized by the model constructed in Scenario 2 is closer to the empirical distribution of liquid $CO_2$ compared to Scenario 1. Meanwhile, the worst-case distribution in Scenario 3 is closest to the empirical distribution of liquid $CO_2$. This indicates that the risk-averse bi-level optimization model based on the P-CVaR model with fractional moments and optimal piecewise number can more reliably estimate the actual demand of liquid $CO_2$. This result is of great significance to obtain more reliable procurement decisions for the Fuhua mine thermodynamic disaster.

4.2.2. Benefits of the Risk-Averse Bi-Level Optimization Model Based on P-CVaR

In the risk-averse bi-level distributionally robust optimization model of emergency material procurement for mine thermodynamic disasters, scenarios with different confidence levels of decision-maker risk-averse preference $\alpha$, equal to 0.75, 0.85 and 0.90, are set to study the impact of the risk-averse size of the decision-maker on the emergency supplies procurement decision. The three confidence levels of $\alpha$ mean that the objective function minimizes the highest 25%, 15%, 5% of all procurement costs under possible random demand. The scenarios settings are as follows:

Scenario 1: Risk-averse bi-level distributionally robust optimization model based on P-CVaR under $\alpha = 0.75$.

Scenario 2: Risk-averse bi-level distributionally robust optimization model based on P-CVaR under $\alpha = 0.85$.

Scenario 3: Risk-averse bi-level distributionally robust optimization model based on P-CVaR under $\alpha = 0.95$.

In the scenarios above, the decision-maker is more risk-averse as the value of $\alpha$ increases. The scenarios above are great methods to verify the impact of decision-makers' risk-averse preference on the emergency supplies procurement decision under different value of $\alpha$.

In terms of solving the optimization models of the above scenarios using the PSO algorithm, the optimal procurement decision of the risk-averse bi-level optimization model for liquid $CO_2$ procurement is shown in Table 6, which includes physical procurement quantity $y = \{y_1, y_2, y_3\}$ and capacity reserve $z = \{z_1, z_2, z_3\}$. The statistical analysis of the optimal decision is shown in Figure 5.

**Table 6.** Optimal procurement decision of the risk-averse bi-level optimization model for liquid $CO_2$ procurements.

| Scenarios | | Scenario 1 | Scenario 2 | Scenario 3 |
|---|---|---|---|---|
| **Risk-Averse Preference** | | $\alpha=0.75$ | $\alpha=0.85$ | $\alpha=0.95$ |
| $y$ | $y_1$ | 83.3676 | 62.1060 | 72.2192 |
| | $y_2$ | 60.0473 | 74.5196 | 56.7818 |
| | $y_3$ | 64.7256 | 71.8019 | 93.1394 |
| | total y | 208.4275 | 208.1405 | 208.4275 |
| $z$ | $z_1$ | 135.9968 | 145.2650 | 224.9239 |
| | $z_2$ | 100.4338 | 131.2661 | 125.3373 |
| | $z_3$ | 166.3836 | 144.5964 | 150.6712 |
| | total z | 402.8142 | 402.8142 | 421.1275 |
| | total | 610.9547 | 629.5550 | 723.0728 |
| | total $y$/total | 34.0680% | 33.1071% | 30.7217% |
| | total $z$/total | 65.9320% | 66.8929% | 69.2783% |

Analyzing the optimal procurement decision of the risk-averse bi-level optimization model for liquid $CO_2$ under different scenarios in Table 6 and Figure 5, the confidence levels of decision-maker risk-averse preference increases from 0.75 to 0.85 and 0.95, and the total of procurement quantity is increased from 610.9547 to 629.5550 and 723.0728. This indicates that the decision-maker risk-averse levels are positively correlated with the total procurement amount of the risk-averse bi-level optimization model. In addition, according to the value of total $z$/total under different scenarios, it can be found that the total purchase volume of production capacity increases more than the actual purchase volume. This indicates that the impact of capacity reserves on averting procurement risk is greater than physical procurement volume. The above content shows that the risk-averse

bi-level distributionally robust optimization model of emergency material procurement for mine thermodynamic disasters is sensitive to risk-averse preference $\alpha$ and capacity reserve quantity $z$.

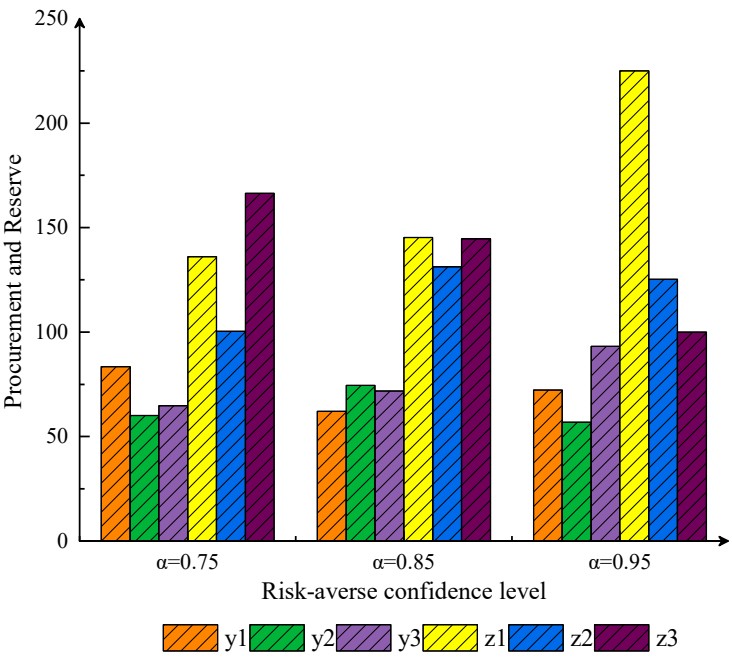

**Figure 5.** The optimal decision of the risk-averse bi-level optimization model for liquid $CO_2$ procurement.

## 5. Conclusions

In this paper, the strategy optimization of emergency material procurement for mine thermodynamic disasters under the joint reserve model is studied. Considering the crucial role of decision-maker risk preferences and reserve models in emergency material procurement, a risk-averse bi-level optimization model is established based on analyzing the game relationship between procurement risks and supplier interests. Although most of the risk-averse optimization models in the literature have only studied the general distribution of emergency material demand, this paper focuses on the situation of small-sample size and heavy-tailed distribution. Considering the crucial role of decision-maker risk preferences and reserve models in emergency material procurement, a P-CVaR distributionally robust optimization model is established to accurately measure the decision-makers' risk preferences when the emergency material demand distribution is small-sample sized and heavy-tailed. Secondly, a risk-averse bi-level optimization model is established to improve the accuracy of averting emergency material procurement risk for mine thermodynamic disasters, based on analyzing the game relationship between procurement risks and supplier in the P-CVR distributionally robust optimization model; this can provide a more accurate risk preference measurement of decision-makers for emergency material procurement than a classical CVaR model. And the total of procurement quantity is increased with the increase in the confidence levels of the decision-maker's risk-averse preference; the total purchase volume of production capacity increases more than the actual purchase volume.

The inspiration for coal mine emergency management can be provided through the emergency material procurement strategy model proposed in this paper, and these management inspirations are as follows. In situations where the emergency supplies' demand for mine thermodynamic disasters supplies is uncertain, the consideration of decision-maker risk preferences should flexibly balance the benefits and risks of emergency material procurement for mine thermodynamic disasters. Coal mines can develop correct reserve and procurement plans based on decision-makers' risk preferences to avoid potential risks, which can improve the emergency management capabilities of thermal and dynamic disasters in mines. The accuracy of the risk-averse preference of decision-makers is crucial

for meeting the actual needs of emergency material. At the same time, in the emergency management preparation stage of coal mines, a reasonable joint reserve model that includes physical reserves and capacity storage should be selected based on demand risk assessment to flexibly meet the needs of emergency situations in mine thermodynamic disasters. This model construction method can improve the efficiency and flexibility of coal mines in the emergency management of thermodynamic disasters.

The performed analysis in this work can be expanded in different aspects. In practical applications, physical reserves and production capacity reserves may still be unable to deal with the uncertainty of emergency material demand. Risk-averse emergency material procurement optimization based on two or more reserve models may be further explored in the future.

**Author Contributions:** Writing—original draft preparation, W.L.; supervision, L.G. All authors have read and agreed to the published version of the manuscript.

**Funding:** National Natural Science Foundation of China (12201275).

**Data Availability Statement:** The research data in this article comes from Table 3.2 of reference [27].

**Conflicts of Interest:** The authors declare no conflict of interest.

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
