# Peer review of "Research on Risk-Averse Procurement Optimization of Emergency Supplies for Mine Thermodynamic Disasters"

_mathematics, doi:10.3390/math12142222_

Round 1
Reviewer 1 Report
Comments and Suggestions for Authors
the paper is good. the author is able to present a good model of optimization. however, the case study is, to some extent, not enough to justify the conclusion as the author only use 1 case study (year 2008). meanwhile, the author picked the case in 2013 to show the urgency of the problem. the author also said there are several cases between 2000 and 2021.
please add another 2-3 cases to strongly justify the conclusion or perhaps provide a better conclusion.
Author Response
“please see the attachment”

Reviewer 2 Report
Comments and Suggestions for Authors
The article discusses a risk-averse procurement model for optimizing emergency supplies in Mining thermodynamics. Please find below my suggestions to improve the quality of the work.
1. The abstract needs to be rewritten coherently to assist the reader in understanding the goal of the study. It is very difficult to understand the goal of the study from these statements “In this study, a novel P-CVaR (Piecewise conditional risk value, P-CVaR) distributionally robust optimization model is proposed to accurately quantify risk preferences. Meanwhile, considering the joint decision of procurement and reserves to improve supply flexibility, a risk-averse bi-level optimization model for pre-disaster emergency material procurement based on the joint reserve model is proposed.” The two sentences seem to be hanging and lack coherence.
2. In section 4.1, the authors assumed lognormal and normal distributions for the empirical distribution of the liquid CO2 demand. The normal distribution is not appropriate as previously stated and obvious from the graph that a positively skewed heavy-tailed distribution is appropriate. Why did the authors consider lognormal as the only heavy-tailed distribution especially when the empirical pdf and cdf showed a lack of fit? A standard procedure should have been to propose several heavy-tailed distributions such as Weibull etc. and compare the goodness of fit of the distributions to the empirical data and then select the best.
3. The scenario design in section 4.2 is not explanatory and it is difficult to follow.
4. The results are not adequately discussed and a comparison of findings to existing literature was not done at all.
Comments on the Quality of English LanguageThe abstract needs to be rewritten coherently to assist the reader in understanding the goal of the study. It is very difficult to understand the goal of the study from these statements “In this study, a novel P-CVaR (Piecewise conditional risk value, P-CVaR) distributionally robust optimization model is proposed to accurately quantify risk preferences. Meanwhile, considering the joint decision of procurement and reserves to improve supply flexibility, a risk-averse bi-level optimization model for pre-disaster emergency material procurement based on the joint reserve model is proposed.” The two sentences seem to be hanging and lack coherence.
Author Response
"Please see the attachment."

Reviewer 3 Report
Comments and Suggestions for Authors
The paper introduces the P-CVaR. Here are some suggestions.
1. The paper lacks the idea behind P-CVaR. 1.1 What are the motivations?
1.2 What are benefits of using this concept?
1.3 What situations that we should adopt this concept?
2. Why do you focus on the fractional moment application? If it is concerned with the problem of small samples, please specify it.
3. Please explain in details of the idea of Eqs. 16 and 27. What is p-bar-L in Eq. 16? If they are not decision variables, how do we specify their values?
4. Please discuss and conclude the issues of piecewise in comparative with the classical models regarding to the result quality.
Round 2
Reviewer 1 Report
Comments and Suggestions for Authors
the paper is good. the author is able to present a good model of optimization. however, the case study is, to some extent, not enough to justify the conclusion as the author only use 1 case study (year 2008). meanwhile, the author picked the case in 2013 to show the urgency of the problem. the author also said there are several cases between 2000 and 2021 -- it is OK now.
please add another 2-3 cases to strongly justify the conclusion or perhaps provide a better conclusion -- it is OK now.
Reviewer 3 Report
Comments and Suggestions for Authors
There is no further suggestions.